# Distribution and Biodiversity of Seed-Borne Pathogenic and Toxigenic Fungi of Maize in Egypt and Their Correlations with Weather Variables

**DOI:** 10.3390/plants11182347

**Published:** 2022-09-08

**Authors:** Yasser M. Shabana, Khalid M. Ghoneem, Younes M. Rashad, Nehal S. Arafat, Bruce D. L. Fitt, Benjamin Richard, Aiming Qi

**Affiliations:** 1Plant Pathology Department, Faculty of Agriculture, Mansoura University, Mansoura 35516, Egypt; 2Department of Seed Pathology Research, Plant Pathology Research Institute, Agricultural Research Center, Giza 12112, Egypt; 3Plant Protection and Biomolecular Diagnosis Department, Arid Lands Cultivation Research Institute, City of Scientific Research and Technological Applications, Alexandria 21500, Egypt; 4School of Life and Medical Sciences, University of Hertfordshire, Hatfield AL10 9AB, Hertfordsire, UK

**Keywords:** maize, seed-borne fungi, weather variables, biodiversity, phylogeny

## Abstract

Studies of the biodiversity of plant pathogenic and toxigenic fungi are attracting great attention to improve the predictability of their epidemics and the development of their control programs. Two hundred maize grain samples were gathered from 25 maize-growing governorates in Egypt and 189 samples were processed for the isolation and identification of seed-borne fungal microbiome. Twenty-six fungal genera comprising 42 species were identified according to their morphological characteristics and ITS DNA sequence analysis. Occurrence and biodiversity indicators of these fungal species were calculated. *Ustilago maydis*, *Alternaria alternata*, *Aspergillus flavus*, *A. niger*, *Penicillium* spp., *Cladosporium* spp. and *Fusarium verticillioides* were the highly frequent (>90% for each), recording the highest relative abundance (˃50%). Al-Menia governorate showed the highest species diversity and richness, followed by Sohag, Al-Nobaria and New Valley governorates. Correlations of 18 fungal species with temperature, relative humidity, precipitation, wind speed, and solar radiation were analyzed using canonical correspondence analysis. Results showed that relative humidity, temperature, and wind speed, respectively, were the most impactful weather variables. However, the occurrence and distribution of these fungi were not clearly grouped into the distinctive climatic regions in which maize crops are grown. Monitoring the occurrence and distribution of the fungal pathogens of maize grains in Egypt will play an important role in predicting their outbreaks and developing appropriate future management strategies. The findings in this study may be useful to other maize-growing countries that have similar climatic conditions.

## 1. Introduction

In Egypt, the area under maize is estimated at 10 million hectares, with a total annual production of 7.5 million tonnes [1]. The main maize-growing governorates are Al-Menia (116,784 ha), Assiut (90,754 ha), Sohag (70,637 ha), Al-Sharkia (97,325 ha), Al-Menofia (78,278 ha) and Al-Behera (77,451 ha) [2].

Maize is attacked by more than 365 pathogens and about 110 diseases [3], mainly caused by fungal pathogens, which lead to a reduction in the quality and quantity of yield and the nutritional value of grains [4] as well as production of mycotoxins that are harmful to humans and animals [5]. Grains of maize represent a good substrate for a large number of fungal pathogens, especially *Fusarium*, *Aspergillus*, *Penicillium*, and *Curvularia* genera, which are the dominant fungi infecting maize. The diversity of phytopathogens on maize grains may be due to the variation in environmental conditions and climatic circumstances, causing distortion, growth diminution, reduction of photosynthetic capacity, and nutrient deficiency as a result of taking nitrogen, carbon, and other inorganic salts from the host [6]. *Fusarium* ear rot, caused by *F. verticillioides* (Sacc.) Nirenberg and *F. proliferatum* (Matsushima) Nirenberg, is a very common damaging disease infecting maize worldwide [7]. It results in a reduction in maize yield by 10% on average, and 30–50% in the severely affected crops. This disease is characterized by grain discoloration and reduction in the quantity and quality of the grain yield [8]. Both fungal species can survive in infected maize grains without noticeable symptoms (by yielding toxins and lytic enzymes) and are then transmitted to developing seedlings, resulting blights and root, stem and ear rot diseases. The pathogen is systemically moved from infected grains to seedlings through the stalk and then the ear [9]. Moreover, mycotoxins, such as fusaric acid, and fumonisins, mainly generated by *F. verticillioides,* have adverse impacts on human, poultry and animal health [5].

Late wilt, or black bundle, is regarded as the most aggressive disease to threaten maize production in Egypt and other maize-growing countries [10], which is caused by the seed- and soil-borne fungus, *Magnaporthiopsis maydis* with synonyms: *Cephalosporium maydis* Samra, Sabet, & Hingorani and *Harpophora maydis* Samra, Sabet and Hingorani) Gams [11]. Leaf spot disease in maize is caused by *C. lunata,* which results in a great loss of maize yield in different regions in China and the United States [12,13]. *Bipolaris maydis*, *Exserohilum rostratum*, and *E. turcicum* were also affirmed to cause severe maize leaf spots [14]. In seedling stage, maize may be attacked by several diseases and many of them are seed-borne. Corn smut (*Ustilago maydis*), *Aspergillus* ear and kernel rot (*A. flavus*), southern corn leaf blight (*B. maydis*), etc. are significant among these diseases [15]. It was found that fungi could be distributed in different types of environments and live in a wide range of temperatures and pHs [16]. Biotic factors, such as the plant host and microbial flora, and abiotic factors, e.g., temperature, humidity, moisture, soil pH, and salinity, have been found to be strongly correlate with the biodiversity and communities of the fungal pathogens and their infection ability [16,17,18]. Human activities, fertilizer constituents, and other agronomic pratices can influence the fungal populations [19]. Genetic markers, such as internal transcribed spacer (ITS) regions, and the highly conserved 5.8S gene are utilized for taxonomic classification and phylogenetic relations among fungal populations [20]. The aim of the present study was to explore the biodiversity and distribution of seed-borne fungi associated with maize grains collected from maize-growing areas all over Egypt. In addition, correlations with different weather variables were also studied to understand the eco-biological relationships of these fungi.

## 2. Materials and Methods

### 2.1. Study Area

A survey of Egypt’s maize-growing governorates was conducted in 2019 (April to August). The area surveyed lies between latitudes 22°16′ N and 33°65′ N, and longitudes 27°87′ E and 33°03′ E, as shown in the map (Figure 1), which was created using ArcGIS software, version 10.1 (Environmental Systems Research Institute (ESRI), 2012). The area surveyed covered 25 governorates with diverse climatic conditions.

### 2.2. Meteorological Conditions

Egypt has generally been affected by an arid to semi-arid desert climate. However, Egypt has four distinct climatic zones: the Mediterranean zone on the northern coast, semi-desertic zone in the middle regions, desertic zone in the southern areas, and red sea climate zone on the eastern coast. Egypt falls in the sub-tropical climatic zone and the tropical zone to the south in upper Egypt. The country is constantly hit by northwesterly wind from the Mediterranean Sea on the northern coast, which leads to moderate temperatures in these areas throughout the year, while the situation changes in the central and the southern parts, where high temperatures occur at night, especially in summer. Every year, from March to May, the country is affected by a strong, hot, dry, and dusty wind which blows from the south or the southwest called *Khamasīn*, causing temporarily rising temperatures, while the humidity levels drop sharply below 5%. Egypt receives between 20 and 200 mm of annual average precipitation along the narrow Mediterranean coast, but it gradually decreases south to Cairo in the central and the southern part of the country (average nearly 0 mm). The annual sunshine duration is high across Egypt, ranging from 3300 h along the northernmost part to reach 4000 h in the interior, in most of the country. In the maize-growing season (April to August, 2019), the mean air temperature ranges between 25.5 and 32.9 °C, the relative humidity between 18.8 and 75.2%, the rainfall from 0 to 0.16 mm day^−1^, the wind speed 4.25 to 5.18 m s^−1^, and the solar radiation 0.67 to 0.73 W m^−2^ [21].

### 2.3. Sampling Process

A total of 200 grain samples of maize were gathered from 25 governorates in Egypt where maize is cultivated, but 189 samples were processed due to samples going missing during transportation, handling, and storage in the lab. Each governorate was represented by 4 districts, each at least 15 km from the next. Two villages in opposite directions with a minimum of five km between them were selected for each district and one maize field was sampled from each village. Subsamples of each sample were randomly collected in a zigzag pattern. The mature corncobs gathered were put in paper bags, numbered, placed in a cool box for transportation, and then stored at 4 °C until tested. For seed health testing of samples, seeds from at least 50 corncobs per sample were extracted and allowed to dry on a bench for 7 days at ambient temperature (25 ± 2 °C). The location was georeferenced for each sampling site using the global positioning system (GPS) and field information was documented. The coordinates were utilized to show the sampling sites on a map (Figure 1) that was generated using ArcGIS software, version 10.1.

### 2.4. Seed Health Testing

The detection of seed-borne mycoflora was performed according to techniques of the International Seed Testing Association [22], including washing and deep-freezing blotter methods.

#### 2.4.1. Deep-Freezing Blotter (DFB) Technique

Maize seed samples (each 400 seeds) were surface-sterilized by soaking in 1% sodium hypochlorite (NaClO) for 3 min, washed by sterile water, and dried on sterilized filter papers at 25 ± 2 °C. For each sample, maize grains were placed on three sterile moistend filter papers in sterile Petri dishes (10-cm diameter) at 10 grains/Petri dish. The dishes were incubated at 20 ± 2 °C for one day and then frozen overnight at −20 °C followed by a 5-day incubation at 20 ± 2 °C under diurnal light provided by two cool white fluorescent lamps. After 7 days’ incubation, the recovered fungi from seeds were carefully examined and identified based on typical morphological characters using a light-supported stereomicroscope at different magnifications (6–60 X). Examination at higher magnifications of compound microscope was performed to confirm the identity of fungal isolates. After purification, the frequency and incidence (mean of % infection, I) for each fungus were calculated using the following equations:(1)$$\mathrm{Frequency}\;\mathrm{of}\;a\;\mathrm{fungus}\;\left(\%\right)=\frac{\mathrm{Number}\;\mathrm{of}\;\mathrm{the}\;\mathrm{fungus}-\mathrm{infected}\;\mathrm{samples}}{\mathrm{Total}\;\mathrm{number}\;\mathrm{of}\;\mathrm{samples}\;\mathrm{tested}}\times 100$$
(2)$$\mathrm{Incidence}\;\mathrm{of}\;a\;\mathrm{fungus}\;\left(\%\right)=\frac{\mathrm{Number}\;\mathrm{of}\;\mathrm{the}\;\mathrm{fungus}-\mathrm{infected}\;\mathrm{grains}}{\mathrm{Total}\;\mathrm{number}\;\mathrm{of}\;\mathrm{grains}\;\mathrm{tested}}\times 100$$

With regard to the smut fungus (*U. maydis*), the density of teliospores was determined (number of teliospores/100 g grains), illustrating the quantity of the spore load in grain sample.

#### 2.4.2. Washing Test

For each sample, 100 g of maize grains were placed in an Erlenmeyer flask containing 100 mL of sterile water with 1 drop of Tween 80. The flasks were shaken for 20 min on an orbital shaker, and then the washing suspension was collected in a beaker and filtered through a piece of cheesecloth. The suspension was centrifuged at 4000 rpm for 15 min and the supernatant was thrown out. The pellet was resuspended by adding one milliliter of sterile water and mixed together with a needle to make an even spore suspension. The obtained suspension was diluted in a known quantity of fluid and examined using a compound microscope. The total number of teliospores in the resulting washing solution was determined using a hemocytometer and expressed per 100 g of grains.

#### 2.4.3. Identification of Seed-Borne Fungi

The fungi obtained were identified based on their cultural, morphological, and microscopic features [7,23,24,25]. Molecular identification was also performed for selected pathogenic and toxigenic fungi. Extraction of their DNA was done according to the guide of the DNA extraction kit (Qiagene, Germany). Amplification was performed using a polymerase chain reaction (PCR) based on the internal transcribed spacer (ITS) region (600 bp) following the procedure prescribed by White et al. [26]. A forward ITS1 primer (5′TCCGTAGGTGAACCTTGCGG3′) and a reverse ITS4 primer (5′TCCTCCGCTTATTGATATGC3′) were used. PCR products were purified using a gel extraction purification kit (Maxim Biotech INC, Rockville, MD, USA) following the manufacturer’s techniques and sequenced (Macrogene Company, Seoul, Korea). The nucleotide sequences were aligned using the ClustalW algorithm and compared with the GenBank database (http://www.ncbi.nlm.nih.gov/BLAST, accessed on 25 February 2022). The DNA nucleotide sequences for the 49 fungal isolates were deposited in GenBank, and the accession numbers of these isolates were attained. MEGA X software version 10.2.4 was used to build the phylogenetic tree based on the maximum likelihood technique with substitution model (Tamura 3-parameter model), rates pattern (Gamma Distributed and Invariant Sites (G + I), and interference options (Nearest-Neighbor-Interchange (NNI), and the number of threads was 2 [27].

### 2.5. Biodiversity Metrics

The biodiversity of the fungal species recovered from maize samples collected from sites allover Egypt was calculated. Frequency (calculated using Equation (1)) and relative abundance (evenness, %) were assessed on a national level utilizing seed health testing data collected across Egypt. The relative abundance was estimated as follows:(3)$$\mathrm{Relative}\;\mathrm{abundance}\;\left(\%\right)=\frac{\mathrm{Number}\;\mathrm{of}\;\mathrm{grains}\;\mathrm{infected}\;\mathrm{with}\;a\;\mathrm{given}\;\mathrm{fungal}\;\mathrm{species}}{\mathrm{Total}\;\mathrm{number}\;\mathrm{of}\;\mathrm{grains}\;\mathrm{infected}\;\mathrm{with}\;\mathrm{all}\;\mathrm{fungal}\;\mathrm{species}\;\mathrm{identified}}\times 100$$

The richness of fungal species and the Shannon–Wiener diversity index (H) were calculated for each of the 25 maize-growing governates. The species richness was determined as the total number of the fungal species identified in a maize-growing governate:

Species richness = total number of fungus species identified in a maize-growing governate.

The Shannon–Wiener diversity index (*H*) was calculated as follows:(4)$$\mathrm{Shannon}–\mathrm{Wiener}\;\mathrm{diversity}\;\mathrm{index}\;\left(H\right)=-\sum_{i=1}^{s}\;P_{i}*Ln\left(P_{i}\right)$$
where *P_i_* = n_i_/N (n_i_ is the number of grains with the species identified I and N is the total number of grains with all fungus species identified), which is the relative abundance demonstrated in a fractional way. *Ln* = is the natural logarithm.

Distribution maps were developed for the occurrence of key pathogenic seed-borne fungi of maize using the software R (2020) and the packages “raster” [28], “sp” [29], and “ggplot2” [30].

### 2.6. Pathogenicity Test

Twenty-two fungal species belonging to six genera were tested for their pathogenicity, as they were the most widespread in this survey. Inoculum of each fungal isolate was prepared by growing them on PDA plates which were incubated at 25 ± 2 °C for one week. Then, 0.5-cm-diameter disks of each fungus were used to inoculate a sterilized medium of sorghum: sand: water (2:1:2 *v*/*v*) that was incubated at 25 ± 2 °C for 15 days.

Pots (40-cm diameter) each containing 6 kg disinfected soil (clay: sand (2:1 *v*/*v*)) and individually infested with the prepared fungal inoculum at a rate of 0.4% (*w*/*w*), and regularly irrigated with tap water to near to field capacity and left for 7 days to achieve the spread of fungal growth. Control pots filled with steam-sterilized soil irrigated by water only.

Healthy maize grains (cv. Monohybrid 168) were surface sterilized by soaking them into a NaCLO solution (1%) for 2 min, then washing with tap water, and allowing to air dry. Six grains were seeded per pot and ten replicates were used for each fungus. A randomized block design was used, and all pots were kept in a greenhouse for 45 days at 31 °C/22 °C day/night temperature and with a 12 h photoperiod. After 14 days, pre- and post-emergence damping-off disease was recorded as percentages, while the percentages of survived plants were recorded 45 days after planting.

### 2.7. Statistical Analyses

The statistical analysis software CoStat version 6.4 (CoHort Software, Pacific Grove, CA, USA) [31] was used for the analysis of variance of the data. Means were compared with Duncan’s multiple range test at *p* ≤ 0.05. The heatmap was used to evaluate the incidence and frequency of the maize seed-borne mycoflora recovered using the TBtools package. Canonical correspondence analysis (CCA) was used to determine the correlations between the incidences of maize seed-borne fungi and weather variables using the software R and the package “vegan” [32]. For each governorate sampled, the weather variables comprised everyday mean air temperature (°C), relative humidity (%), and wind speed (km h^−1^), as well as the monthly average precipitation (mm) and solar radiation (kWh m^−2^) from April to August 2019.

## 3. Results

### 3.1. Distribution of Maize Seed-Borne Mycobiota

Among 200 samples of maize grains gathered from 25 maize-growing governorates of Egypt in the 2019 cropping season, this study only considered 189 samples due to the loss of 11 samples during transportation. The maize samples collected represented thirteen hybrids of the Egyptian maize cultivars which were grown throughout the sampled maize-growing fields, including yellow maize hybrids (SC-168, SC-178, SC P3433, TWC-360 and TWC-368) and white maize hybrids (SC-30 K8, SC-30 K9, SC-132, SC-2030, SC-2031, SC-30P74, TWC-321 and TWC-324).

Forty-two species from 26 genera of fungi were detected on the grains gathered. Using the DFB method, 25 genera and 41 fungal species were isolated from maize grains, while one smut fungus was recovered using the washing technique (Figure 2 and Figure 3). Among them, *U. maydis*, *F. verticillioides*, *A. niger*, *Penicillium* spp., and *A. flavus* had the greatest average frequency (˃85%) and incidence (˃33%) in all the maize-growing governorates. Geographically, densities of *U. maydis* teliospore were greater in the southern governorates across the Nile valley, particularly Aswan and Qena, and in Northern-east governorates, especially in Port Said and Sinai (Figure 2). *Cladosporium* spp. and *A. alternata* came next with a high average frequency (60.7 and 57.4%, respectively) and incidence (17.73 and 4.98%, respectively). *Acremonium* spp. and *F. incarnatum* recorded a lower frequency (30.2 and 30.1%, respectively), followed by *Fusarium* spp. and *A. terreus* (24.7 and 19.6%, respectively), while *F. proliferatum*, *Trichothecium roseum*, *Sarocladium zeae*, and *Nigrospora* spp. were less frequently recorded (15.5, 14, 13.5 and 12.1%, respectively). The remaining fungi with an average frequency < 10% were found only in a few governorates.

### 3.2. Biodiversity of Maize Seed-Borne Fungi

Regarding the total frequency and relative abundance analysis of the seed-borne fungi isolated, seven fungal species showed high frequency (92–100%) and relative abundance (58.6–61.0%), namely *U. maydis*, *A. alternata*, *A. flavus*, *A. niger*, *Penicillium* spp., *Cladosporium* spp., and *F. verticillioides*. *Acremonium* spp., *F. solani*, and *Fusarium* spp. came in the second order with frequency percentages ranging from 68% to 72% and relative abundance from 41.5% to 44.0%. *Aspergillus ochraceus,*
*B. maydis,*
*E. rostratum*, *G. candidum*, *Nigrospora* spp., *Stemphylium* sp., and *T. roseum* came third with moderate percentages of the total frequency from 32% to 56% and relative abundance from 19.51% to 34.15%. The other fungal species occurred at lower frequencies (12–28%) and relative abundances (7.14–17.07%). The seed-borne fungi *A. flavus*, *F. verticillioides*, *Penicillium* spp., and *U. maydis* were the most frequent (detected in 100% of samples collected) and most abundant (detected in 61.0% of grains tested) (Figure 4).

The seed-borne fungi associated with maize samples differed among the 25 governorates surveyed. The species diversity and richness of the seed-borne fungi obtained from maize-growing governorates are shown in Figure 5. Al-Menia governorate recorded the highest species diversity (3.01), followed by Sohag and Al-Nobaria (2.89 and 2.81, respectively), while the lowest species diversity (2.18) was recorded for Al-Ismaelia. Moreover, the greatest species richness was in Al-Menia governorate (27 species), followed by Sohag, Al-Nobaria, and New Valley governorates (24, 22, and 21 species, respectively), while the least species richness was observed in Alexandria and Al-Ismaelia governorates (10 species for each). The distribution and incidence or density of the key pathogenic seed-borne fungi of maize were mapped geographically (Figure 6 and Figure 7). The three main fungi with a high incidence (≥91%) were *U. maydis*, *F. verticillioides* and *A. niger* with a greater incidence in the northern areas and Nile Delta. *Penicillium* spp. And *A. flavus* were the fourth and fifth most common pathogens (with an average incidence 89.7 and 85%, respectively) in the northern areas and Nile Delta. *Cladosporium* spp. And *A. alternata* were the most dominant pathogens present in 23 and 24 governorates (60.7% and 57.4%, respectively) and mainly in the Nile Delta. *Acremonium* spp. was present in only 18 governorates and mainly in Al-Sharkia and Aswan governorates. *Fusarium incarnatum* and *F. proliferatum* pathogens were detected in only 13 and seven of the governorates, respectively, with the greatest incidence in Luxor, Kafr El-Shekh and Sohage, and Al-Qalyobia governorates.

### 3.3. Phylogenetic Analysis

A phylogenetic tree illustrated in Figure 8 shows the phylogenetic relationships amongst the fungi isolated. Isolates displayed <97% similarity with the formerly comparable isolates referenced in GenBank. The accession numbers of the 49 pathogenic fungi that were obtained from the GenBank for *Alternaria alternata* (EG1M1-1), *Bipolaris maydis* (EG2M1-1), *Curvularia hawaiiensis* (EG3M1-1, EG3M1-2), *Curvularia lunata* (EG3M2-1), *Curvularia tsudae* (EG3M3-1, EG3M3-2, EG3M3-3), *Exserohilum rostratum* (EG4M1-1, EG4M1-2, EG4M1-3, EG4M1-4, EG4M1-5, EG4M1-6), *Fusarium chlamydosporum* (EG5M1-1, EG5M1-2), *Fusarium fujikuroi* (EG5M2-1), *Fusarium incarnatum* (EG5M3-1, EG5M3-2, EG5M3-3, EG5M3-4), *Fusarium proliferatum* (EG5M4-1, EG5M4-2, EG5M4-3), *Fusarium verticillioides* (EG5M5-1, EG5M5-2, EG5M5-3, EG5M5-4, EG5M5-5, EG5M5-6, EG5M5-7, EG5M5-8, EG5M5-9, EG5M5-10, EG5M5-11, EG5M5-12, EG5M5-13, EG5M5-14, EG5M5-15, EG5M5-16, EG5M5-17, EG5M5-18, EG5M5-19, EG5M5-20), *Sarocladium implicatum* (EG6M1-1), and *Sarocladium zeae* (EG6M2-1, EG6M2-2, EG6M2-3, EG6M2-4). The tree classified the fungal species into three clades. The first clade contains three strains of *F. incarnatum* (EG5M3-1, EG5M3-2 & EG5M3-4), one strain of *F. fujkuroi* (EG5M2-1) had support values (94% BP), and six strains of *E. rostratum* (EG4M1-1, EG4M1-2, EG4M1-3, EG4M1-4, EG4M1-5& EG4M1-6) with (89% BP). The second clade contains four different *Fusarium* species, including twenty strains of *Fusarium verticillioides* (EG5M5-1, EG5M5-2, EG5M5-3, EG5M5-4, EG5M5-5, EG5M5-6, EG5M5-7, EG5M5-8, EG5M5-9, EG5M5-10, EG5M5-11, EG5M5-12, EG5M5-13, EG5M5-14, EG5M5-15, EG5M5-16, EG5M5-17, EG5M5-18, EG5M5-19 & EG5M5-20) that had support values (88–99% BP), three strains of *F. proliferatum* (EG5M4-1, EG5M4-2 & EG5M4-3), two strains of *F. chlamydosporum* (EG5M1-1& EG5M1-2), and one strain of *F. incarnatum* (EG5M3-3). The third clade was also classified into two sub-clades; the first sub-clade with (97% BP), including two strains of *C. hawaaiiensis* (EG3M1-1 and EG3M1-2), three strains of *C. tsudae* (EG3M3-1, EG3M3-2, EG3M3-3), and one strain of *C. lunata* (EG3M1-4), whereas the second sub-clade with (90% BP) contains four strains of *S. zae* (EG6M2-1, EG6M2-2, EG6M2-3& EG6M2-4), one strain of *S. implicatum* (EG6M1-1), and one strain of *A. alternata* (EG1M1-1).

### 3.4. Pathogenicity Test

A total of twenty-two fungal species comprising six genera recovered from maize seeds were selected for pathogenicity assays to explore their capacity to infect and kill maize seedlings (Table 1). Results indicated that all fungi tested were pathogenic to maize seeds/seedlings with varying degrees of pathogenicity. In this regard, *F. incarnatum* (isolates EG5M3-1) was the most virulent (only 45% of seedlings survived, Table 1). *E. rostratum* (isolate EG4M1-2) and *F. incarnatum* (isolate EG5M3-3) were the second most virulent isolates (51.7% survival, for both), followed by *F. verticillioides* (EG5M5-2), *F. nygami* (EG5M6-1), *E. rostratum* (EG4M1-1), *B. maydis* (EG2M1-2), *S. zeae* (EG10M2-2), and *B. maydis* (EG2M1-2) (57.1%, 57.1%, 57.8%, 58.3%, 59.8%, and 60.3% survival, respectively) (Table 1). The least virulent isolates were *Alternaria alternata* and *Cephalosporium acremonium* (86.5% and 84.4% survival, respectively) (Table 1).

The disease symptoms included elongate and elliptical lesions on leaves, which later merged or coalesced and blighted the entire leaf. Infected seedlings wilted and died within 3–4 weeks. Yellowing and stunting due to infection were also observed. Among four *Sorocladium zeae* strains tested, isolate EG10M2-2 showed the greatest percentages of rotted seeds (28.4%) and seedling mortality (11.8%). Affected plants showed wilting of uppermost leaves and a browning vascular system in the lower portion of stems. Irregulated vascular strands due to pathogen invasion were noticed, which was accompanied by weakness of the stalk. Symptoms continued to affect the leaves showing chlorosis, necrosis, and yellowing which afterward turned into black. The growing-on test showed similar symptoms in all *Fusarium* species treatments; rotted grains, stunted and yellow seedlings. Infection with *F. proliferatum* (isolate EG5M4-3), followed by *F. verticillioides* (isolate EG5M5-1), *S. zeae* (isolate EG6M2-2), *F. proliferatum* (isolate EG5M4-2), *F. verticillioides* (isolates EG5M5-2 and EG5M5-3) and *F. incarnatum* (isolates EG5M3-1 and EG5M3-3) gave the greatest percentages of rotted grains (33.1, 29.8, 28.4, 26.6, 25.9, 25.5, 25.0 and 25.0%, respectively). Four weeks after inoculation, most fungi tested caused mild to severe disease on maize seedlings. *F. incarnatum* (isolate EG5M3-1), followed by *E. rostratum* (isolate EG4M1-2), *F. incarnatum* (isolate EG5M3-3), *F. nygamai* (isolate EG5M5-1), *B. maydis* (isolate EG2M1-3), and *E. rostratum* (isolate EG4M1-1) caused seedling mortality of 30.0%, 25.2%, 23.3%, 22.9%, 21.7%, and 21.2%, respectively. In the case of *Fusarium* infection, white fluffy colonies were observed on the grains and around the base of seedlings. Mild disease occurred on maize grains and seedlings grown on soil infested with *A. alternata*, *C. acremonium,* or *F. chlamydosporum* isolates when compared with the control.

### 3.5. Correlations between the Occurrence of Maize Fungal Pathogens and Weather Variables

The correlations between the occurrence of the seed-borne fungi obtained and the weather variables were investigated. A negative correlation was observed between temperature and relative humidity (*r* = −0.91, *p* ≤ 0.001), whereas temperature was positively correlated with wind speed (*r* = 0.15, *p* ≤ 0.05) and solar radiation (*r* = 0.60, *p* ≤ 0.001, Table 2). Solar radiation was negatively associated with the relative humidity (*r* = −0.70, *p* ≤ 0.05) but positively associated with wind speed (*r* = 0.18, *p* ≤ 0.05). A negative relationship was also noticed between precipitation and relative humidity (r = −0.22, *p* ≤ 0.05) (Table 2). A canonical correspondence analysis (CCA) was performed to reveal the correlations between the occurrence of maize seed-borne fungi and these five weather variables (Figure 9). The first three axes explained 93% of the species variance in the dataset (Table 3). In Figure 9, the positions of fungal species are expressed as red triangles and the weather variables are expressed as arrows, where the length of the arrow denotes the effectiveness of the weather variable in the data interpretation and the direction of the arrow refers to the greatest change in the weather variable. Results from CCA indicated that relative humidity (with a good correlation with the two axes), temperature (mainly associated with axis 2), and wind speed (mainly correlated with axis 1) were the most influential weather variables. Solar radiation and precipitation were less influential with solar radiation correlating with both axes while precipitation mainly correlated with axis1.

The orthogonal projections of *T. roseum* and *F. incarnatum* on the arrows for temperature, solar radiation, and precipitation showed that they were the only two pathogens with a high requirement for these environmental variables compared to the other pathogens (Figure 9). *Cladosporium* spp. And *A. tamari* showed smaller requirements for temperature and solar radiation than *T. roseum* and *F. incarnatum* and below average requirements for precipitation and relative humidity. *A. tamari* and *Sarocladium zeae* were the two pathogens with a high requirement for wind speed. *Aspergillus* pathogens (including *A. niger* and *A. flavus*, the second and third pathogens with the highest average incidences in Egypt, respectively) showed a low positive correlation with temperature, solar radiation, and wind speed, and lower than average requirement for relative humidity and precipitation. The other fungal pathogens, e.g., *Stemphylium* spp., *Acremonium* spp., *Penicillium* spp. (the pathogen with the highest average incidence in Egypt), and *F. verticillioides* (fourth highest), showed positive requirements for relative humidity but negative correlations with temperature and solar radiation. The site representations on the di-plot did not show any distinctive group related to climatic areas (data not shown).

## 4. Discussion

This work represents the first comperhensive monitoring of maize pathogenic seed-borne fungi all over Egypt. Some small-scale reports have described the presence and distribution of some of these fungal pathogens, but they have given only a limited vision of the risks relevant to the infestation of these pathogens. In addition, we also investigated correlations of different weather conditions with biodiversity and the distribution of different fungi associated with maize grains. Our findings revealed that 42 fungal species containing pathogenic, saprophytic, and toxigenic types were recorded in varying extents. Results obtained showed that maize grains were accompanied by several fungal pathogens with high frequency (>90%) and relative abundance (>50%) across the 25 governorates, namely *U. maydis*, *A. alternata*, *A. flavus*, *A. niger*, *Penicillium* spp., *Cladosporium* spp., and *F. verticillioides*, which are the main causal agents of corn common smut, Alternaria leaf blight, and different ear rots diseases in the majority of world maize-growing areas [33,34,35,36]. Common smut of maize, caused by *U. maydis,* has a worldwide distribution, leading to high economic losses that may reach to 10% in susceptible sweet corn hybrids [37]. *Fusarium proliferatum* and *F. verticillioides* are the most devastating predominant causal agents of Fusarium ear rot disease of maize worldwide [7], which typically leads to a reduction in maize yield by 10% and by 30–50% in the severely affected crops [9]. The accumulation of mycotoxins, such as deoxynivalenol, zearalenol, and fumonisin, in the pre-harvest infected plants or in stored grains due to Fusarium ear rot pathogens has harmful effects on human health, poultry, and animals [34,35]. The present study showed that *A. niger*, *A. flavus* and *Penicillium* spp. were the most dominant post-harvest pathogenic fungi. Association of maize grains with *Aspergillus* and *Penicillium* species has been reported in stored grains, as well as in maize plants in the field [38] causing great economic loss by lowering quality of the infected maize grains. Furthermore, health-hazardous mycotoxins, such as aflatoxins, ochratoxin, and patulin, are also commonly associated with their presence [39]. Several fungal species of the genera *Cladosporium*, *Alternaria,* and *Epicoccum* were often isolated from maize grains in this study. Contamination by these pathogens is common in maize grains causing black point disease and reducing their market quality, and producing hazard mycotoxins [40,41]. A group of mycotoxins produced by *Alternaria* spp. has been reported, including tetramic acid derivatives, tenuazonic acid, dibenzopyrone derivatives, alternariol, alternariol mono-methyl ether, and altenuene and perylene derivatives, altertoxins [42], which can prevent the germination of maize as well as other vegetable and crop seeds [43].

The diversity of fungal microbiomes in each of the maize-growing governorate was studied using Shannon–Wiener diversity index. The Shannon diversity index is a popular measure used in ecology to assess the diversity of species in a population. The index takes into consideration the number of species present in a habitat (richness) and their relative abundance (evenness) [44]. So, the dissemination level of fungi among the species and their abundance in the governorates could be monitored.

Biodiversity data in our study indicated that Al-Menia governorate has the greatest species diversity and richness, followed by Sohag, Al-Nobaria, and New Valley governorates. In this regard, *U. maydis*, *F. verticillioides,* and *A. niger* pathogens showed the highest incidence, followed by *Penicillium* spp. and *A. flavus*. This finding is fully in line with the results obtained by Goko et al. [45] and Elwakil et al. [33] who reported that *F. verticillioides* and *Aspergillus* species, e.g., *A. flavus* and *A. niger*, were more frequent pathogens in the cultivated-maize fields. The biodiversity and species abundance in such regions may be attributed to the semi-arid to arid climates associated with elevated temperature and average humidity and wind speed levels, which might be suitable for the germination and dessimination of a wide range of fungal spores.

Phylogenetic relationships among 49 fungal isolates within six genera were verified in the current study. The major groups and their branched clusters showed high bootstrap values (>97%) denoting highly significant relations between members of each cluster/clade. The maximum likelihood analysis depending on ITS sequences revealed that *Curvularia, Alternaria,* and *Bipolaris* species are clustered in one clade, in which *A. alternata* and *C. lunata* are discretely clustered in a sub-clade and *B. maydis* is assembled in another sub-clade. Our findings revealed that both *E. rostratum* and *B. maydis* are closely linked to *Curvularia* spp., but phylogenetically they were different from each other. This finding is fully in line with the finding of Kirk et al. [46] and Manamgoda et al. [46], who reported that *Curvularia*, *Exserohilum,* and *Bipolaris species* are closely related to each other with same teleomorph (*Cochliobolus*). In this connection, Manamgoda et al. [47,48], based on joint genetic analysis of rDNA ITS, 28S, GAPDH, and translation elongation factor 1-α genes, re-evaluated the taxonomy of the genera *Bipolaris*, *Cochliobolus*, and *Curvularia* into two monophyletic groups: *Bipolaris* and *Cochliobolus* species clustered in group 1 combined with their respective type species *B. maydis* (Y. Nisik. & C. Miyake) Shoemaker. In the contrary, *Curvularia* species that have the former names *Bipolaris*, *Cochliobolus*, *Pseudocochliobolus* and recently re-divided as *Curvularia* were clustered in group 2 with its generic type *C. lunata*. Some species of *Bipolaris* genus have been re-divided into genus *Curvularia*, comprising *C. australiensis*, *C. coisis*, *C. ellisii*, *C. graminicola*, *C. hawaiiensis*, *C. ovariicola*, *C. spicifera*, *C. ravenelli*, and *C. tripogonis* [49]. The second major group in our phylogenetic tree comprises four species within the genus *Fusarium*. Even though the four detected *Fusarium* species fall into a monophyletic sub-cluster, they were affiliated with three distinct sections, i.e., Arthrosporiella (*F. incarnatum*), Sporothrichiella (*F. chlamydosporum*), and section Liseola and their allies (*F. fujikuroi*, *F. proliferatum,* and *F. verticillioides*). The most dominant species was *F. verticillioides* (20 isolates; 42.6%) which belonged to the section Liseola. This species was the most prevalent species associated with the maize; being present in 93.0% of the maize samples. In this connection, subsequent molecular studies showed that section Liseola was paraphyletic [50]. Our findings are in line with that reported by Fallahi et al. [51] on *Fusarium* species isolated from maize seeds, which might clarify the sub-grouping of the five *Fusarium* species into diverse clades in the identical sub-cluster. The classification of the *Fusarium* genus is complex and recently includes nearly 1000 species have been recognized and divided into 16 sections, with approaches differing between wide and narrow conceptions of speciation [52,53]. Based on the current phylogenetic analyses of the rDNA cluster and the genes of *β-tub*, *EF-1α*, and *lys2*, the taxonomy of *Fusarium* species was confirmed to comprise seven major clades, I–VII. In conclusion, several divisional relationships among *Fusarium* spp. remain fuzzy [53]. Our phylogenetic analysis revealed that the second major sub-group in our phylogenetic tree contained another distinct and well-supported clade, which comprised four *S. zeae* (synonym *Acremonium*
*zeae* Gams & Sumner) in one subclade and *S. implicatum* in the other subclade. The genus *Acremonium* is a complex and big polyphyletic genus of Ascomycota, containing about 150 species, and littering in different orders of Sordariomycetes [54]. According to a new DNA phylogenetic study, several species of *Acremonium* (Hypocreales) have been revised and it was moved to genus *Sarocladium,* but the relationship between both genera remains unclear [53]. Despite the morphological similarity of both genera and members of the order Hypocreales, they are phylogenetically differed, of which the species of type *Acremonium* is belonged to *Bionectriaceae* while that of *Sarocladium* is still believed as incertae sedis [55]. Although, *Acremonium zeae* (*Sarocladium zeae*) was previously described as the seed-borne causal pathogen of black-bundle disease of maize [56], current studies reported *S. zeae* as a useful endophyte in maize seeds collected from different countries [55,57,58]. Our pathogenicity experimental data demonstrated the potentiality of *S. zeae* (EG6M2-1, EG6M2-2, EG6M2-3 and EG6M2-4) pathogens to cause symptoms on maize seedlings in the form of a great increase of seedling mortality in comparison to the healthy control.

The ordination diagram of CCA indicated that relative humidity, temperature, and wind speed were the major effective weather variables, followed by solar radiation and precipitation. Humidity is a crucial determining factor for fungus vitality influencing their development, multiplication and pathogenicity in many cases. Some of these fungi prefer high humidity to grow, while others fit medium moisture levels, and some have a tendency to grow better below low moisture levels. Generally, high-moisture content may be essential in some fungal species for spore germination, as it is needed for starting the germination. This finding corresponds with that of Pfordt et al. [39], who reported that temperature and relative humidity are significant metrological variables influencing the range of *Fusarium* spp. of ear and stalk rot infection of maize. In this regard, over four years of CCA studies showed the importance of relative humidity on the aeromycoflora, mainly on spores of Ascomycetes and Basidiomycetes [59]. In many Ascomycetes, humidity is important to exert a high osmotic pressure within the fungal ascus to release the ascospores into the atmosphere [60]. In most species of Basidiomycota, humidity is necessary for discharge of basidiospores, which operate by the rapid movement of a droplet of fluid, called Buller’s drop, over the surface of spore [61].

Results from the CCA revealed a strong relationship of incidence of the two pathogenic fungi *T. roseum* and *F. incarnatum* with air temperature, solar radiation and precipitations than the others. However, *Cladosporium* spp. and *A. tamari* showed closer correlation with temperature and solar radiation than the two previously mentioned pathogens, and below their average needs for both precipitation and relative humidity variables. These findings are in agreement with those obtained by Grinn-Gofron and Bosiacka [60] who reported the positive correlation of *Cladosporium*, *Alternaria*, *Drechslera* type, *Ganoderma* and *Epicoccum* spores with the air temperature. However, *Cladosporium*, and *Drechslera* type showed low average requirements for the wind speed variable [59]. Stress induced by temperature level can affect the dynamics of host/pathogen interactions and ultimately results in changes in the virulence of the fungus. These changes in behavior of phytopathogens can be attributed to the temperature effects on their fungal growth by influencing mobility of the cellular enzymatic reactions and changing secretome of the cell [62]. Results from the CCA showed a high correlation of incidence of *A. tamari* and *S. zeae* pathogens with the wind speed variable. Wind is one of the critical factors influencing the release and dispersal of many air-borne spores, which varies according to location and season [59]. In this concern, Lin and Li [63] reported a high negative correlation between fungal spore concentration and wind speed when the wind speed was below 5 m s^−1^. Other fungi, such as *Stemphylium*, *Penicillium*, *Acremonium,* and *F. verticillioides,* showed a positive correlation with relative humidity and a negative correlation with temperature and solar radiation. *Aspergillus niger* and *A. flavus* showed low positive correlation with temperature, solar radiation, and wind speed and lower correlation with relative humidity and precipitations. *Aspergillus flavus* and *F. verticillioides* have the capability to grow and sporulate on a broade range of temperature levels but the optimum growth temperature for *A. flavus* is 30 °C [64], whereas *F. verticillioides* is favored by lower temperatures ranging between 20 and 25 °C [65]. This finding is in agreement with that obtained by Lanubile et al. [66], who recorded the highest induction of two pathogenesis-related (PR) FUM genes with aflatoxin production ranging from 25 to 30 °C in the maize grains singly affected by *F. verticillioides* or *A. flavus* infection.

## 5. Conclusions

In conclusion, the present study provided background data on the biodiversity, pathogenicity, phylogenetic relationships, and distribution of important pathogenic and toxigenic seed-borne fungal microbiota affecting maize in Egypt. Moreover, it clarified the correlations between occurrence of these fungi and various weather variables. Forty-two fungal species were identified comprising saprophytic and pathogenic ones, some of them important mycotoxin producers. In this connection, relative humidity and temperature were the most influential metrological variables. An ordination diagram of CAA analysis showed that the occurrence and distribution of the studied fungi in maize grain samples did not show a clear climatic distribution in the studied area. Our results reported the broad incidence and distribution of seed-borne and toxigenic pathogens across different Egyptian maize-cropping sites. The presence and prevalence of such pathogens draw attention to the importance of developing protection strategies to improve food security and safety. Monitoring of these pathogens may strongly help in the early warning to avoid their outbreaks and the resultant economic loss. Our findings can be helpful for other maize-growing countries that practice the same climatic conditions.

## Figures and Tables

**Figure 1 plants-11-02347-f001:**
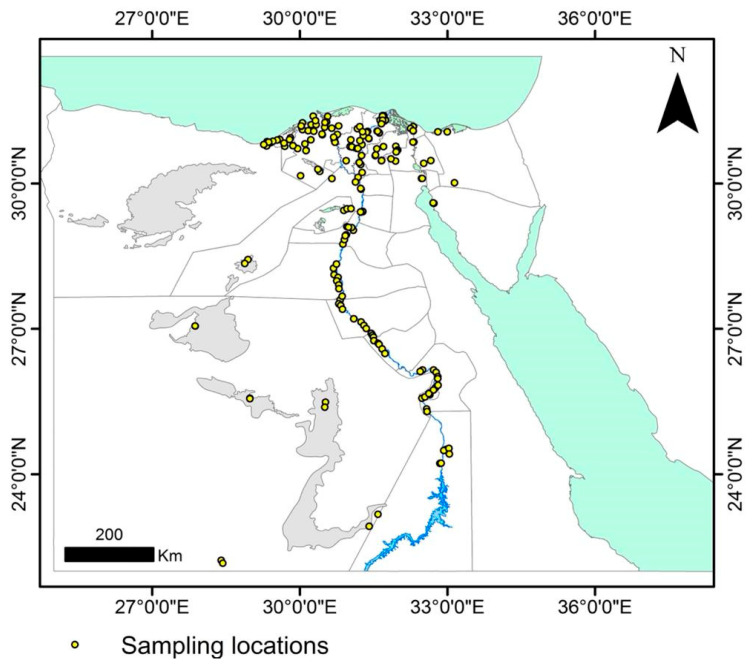
Locations (yellow dots) where maize grains were sampled from 25 maize-growing governorates of Egypt.

**Figure 2 plants-11-02347-f002:**
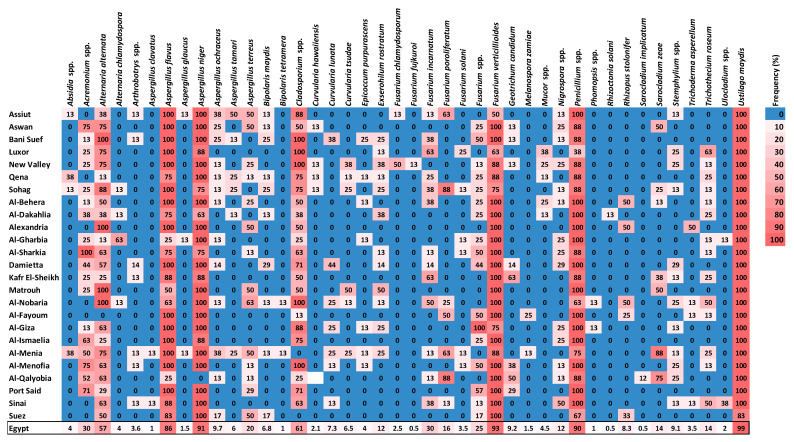
Heatmap showing frequencies of 42 maize seed-borne fungus species in each of the 25 maize-growing governorates of Egypt and for the national average over Egypt. Gradients of frequency key on the right vary from 0 (blue) to 100% (red).

**Figure 3 plants-11-02347-f003:**
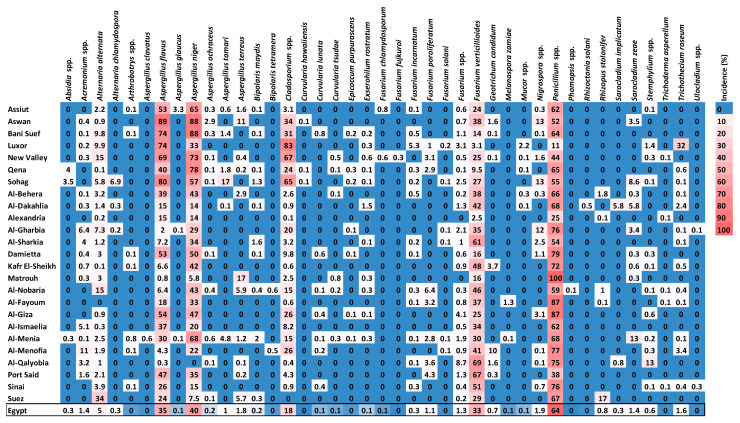
Heatmap showing average incidences (% grains affected) of 41 maize seed-borne fungus species in each of the 25 maize-growing governorates of Egypt and for the national average over Egypt. Gradients of frequency key on the right vary from 0 (blue) to 100% (red).

**Figure 4 plants-11-02347-f004:**
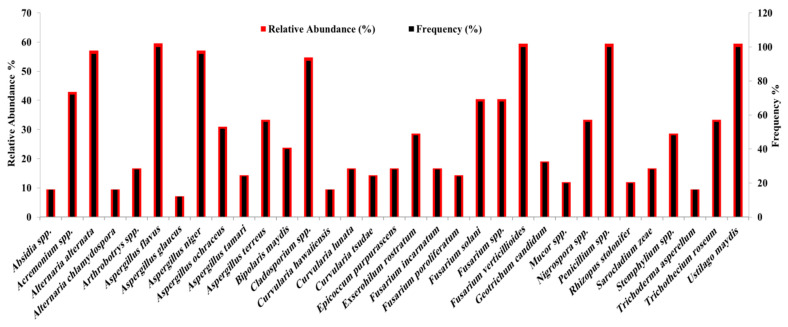
Relative abundance (%) and frequency (%) of maize seed-borne fungi detected using the deep-freezing blotter technique and one smut fungus (*Ustilago maydis*) detected with the washing method calculated from data aggregated from all 25 maize-growing governorates of Egypt.

**Figure 5 plants-11-02347-f005:**
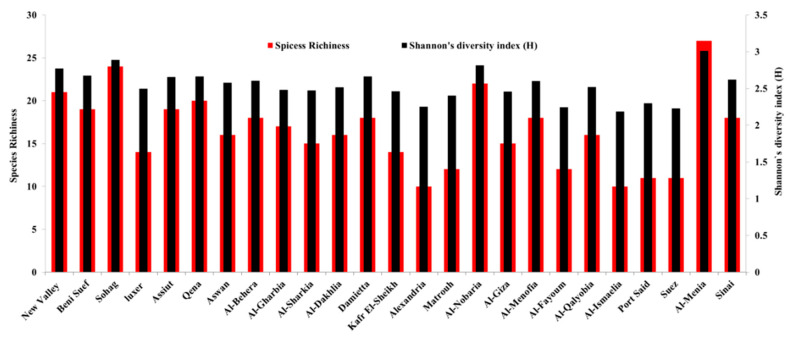
Species richness (total number of fungus species identified) and species diversity (the Shannon-Wiener diversity index) calculated from the seed-borne fungi recovered from maize grain samples in each of the 25 maize-cropping governorates in Egypt.

**Figure 6 plants-11-02347-f006:**
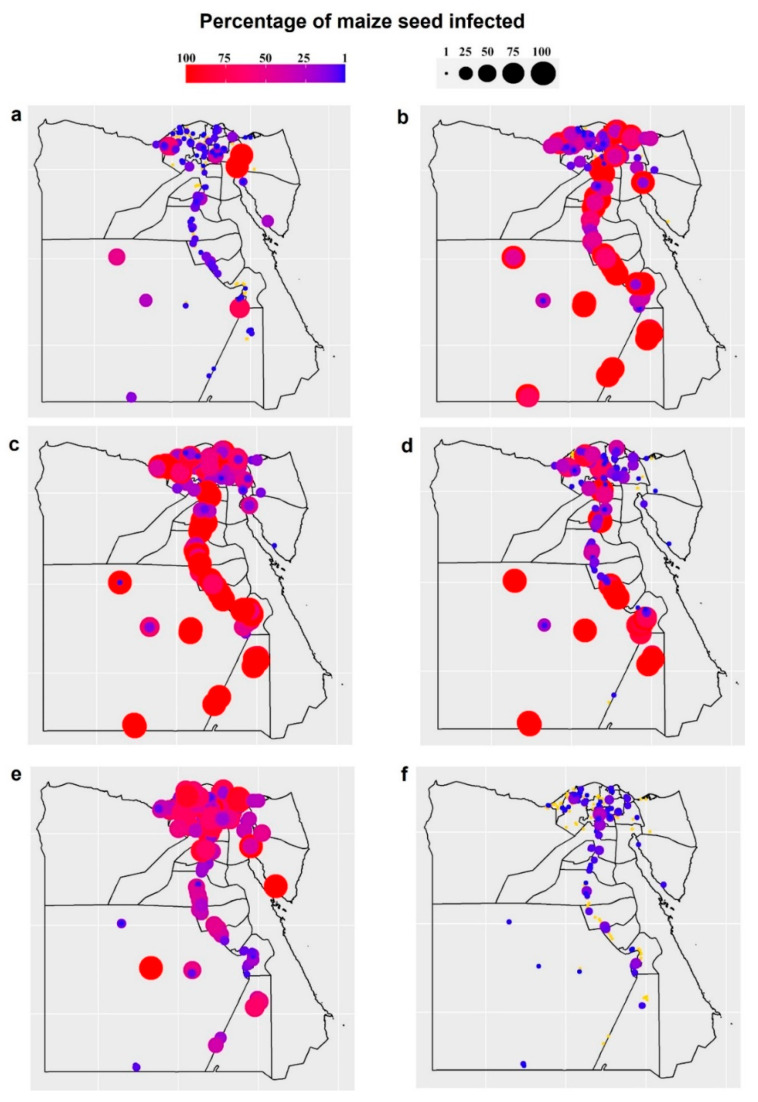
Geographical distribution and incidence (%) of key maize seed-borne fungi recovered from grain samples collected from the 25 maize-growing governorates in Egypt, where (**a**) *Alternaria alternata* (**b**) *Aspergillus flavus*, (**c**) *A. niger*, (**d**) *Cladosporium* spp., (**e**) *Fusarium verticillioides*, (**f**) *Penicillium* spp. (circles in yellow indicate no occurrence of the pathogen).

**Figure 7 plants-11-02347-f007:**
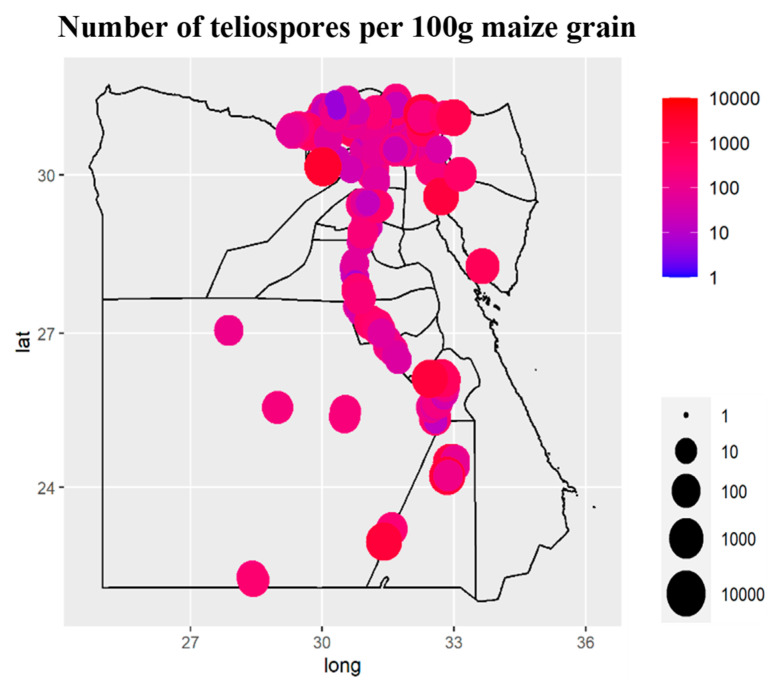
Geographical distribution and density (number of teliospores per 100 g grain) of common smut fungus (*Ustilago maydis*) in maize grain samples collected from the 25 maize-cropping governorates in Egypt. Circles in yellow indicate no occurrence of the pathogen.

**Figure 8 plants-11-02347-f008:**
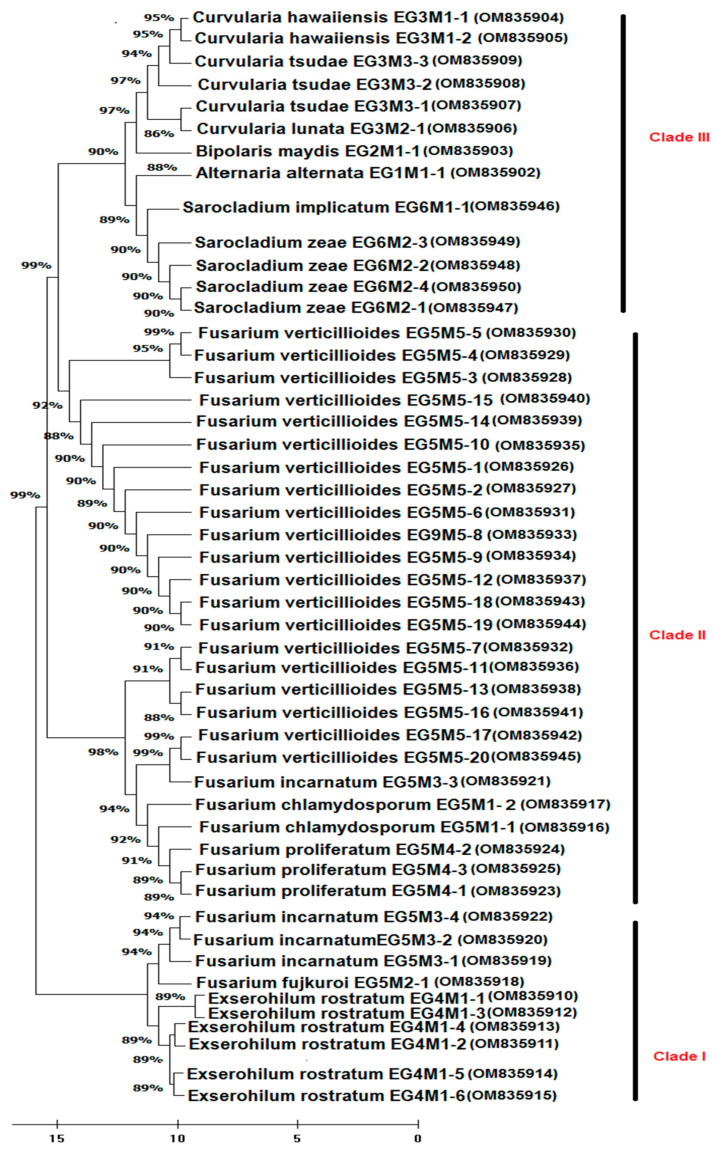
Phylogenetic tree of the seed-borne fungal species [scientific name, isolate code, (GenBank accession number)] isolated from maize grain samples in all 25 maize-cropping governorates of Egypt.

**Figure 9 plants-11-02347-f009:**
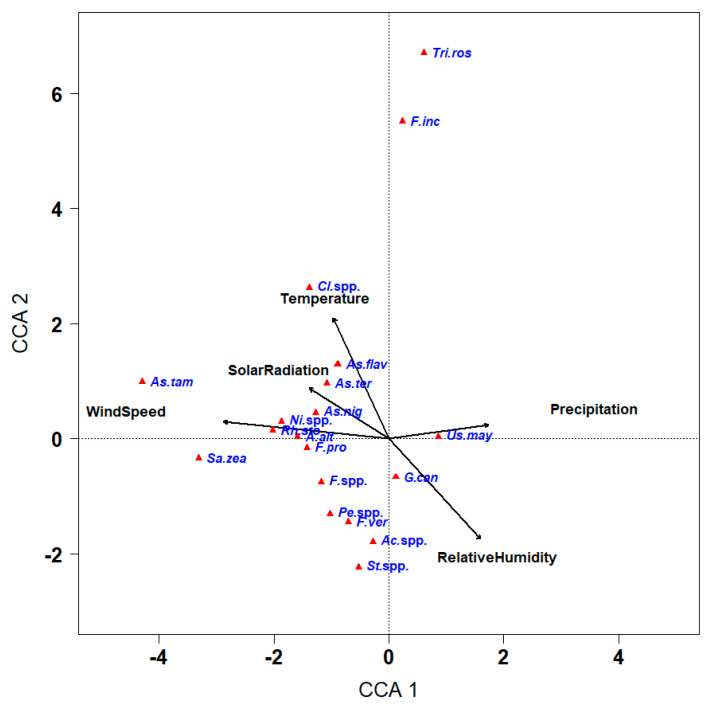
Ordination diagram of canonical correspondence analysis of seed-borne mycoflora in maize grain samples collected from 25 maize-growing governorates with five weather variables. The red triangles denote the fungal pathogens and line arrows denote the weather variables. *Ac.* spp. = *Acremonium* spp., *A. alt* = *Alternaria alternata*, *As. flav* = *Aspergillus flavus*, *As. nig* = *A. niger*, *As. tam* = *A. tamari*, *As. ter* = *A. terreus*, *Cl.* spp. = *Cladosporium* spp., *F. inc* = *Fusarium incarnatum*, *F. pro* = *F. poroliferatum*, *F.* spp. = *Fusarium* spp., *F. ver* = *F. verticillioides*, *G. can* = *Geotrichum candidum*, *Ni.* spp. = *Nigrospora* spp., *Pe.* spp. = *Penicillium* spp., *Rh*. sto = *Rhizopus stolonifer*, *Sa. zea* = *Sarocladium zeae*, *St.* spp. = *Stemphylium* spp., *Tri. ros* = *Trichothecium roseum*, *Us. may* = *Ustilago maydis*.

**Table 1 plants-11-02347-t001:** Pathogenicity against maize of seed-borne fungi isolated.

Fungus	Code	Pre-Emergence Damping Off (%)	Post-Emergence Damping Off (%)	Survival (%)
*Alternaria alternata*	EG1M1-1	6.8 l	6.7 ij	86.5 b
*Bipolaris maydis*	EG2M1-1	23.4 ef	13.4 f	63.2 hi
	EG2M1-2	22.3 f–h	17.3 e	60.4 i–k
	EG2M1-3	20.0 hi	21.7 cd	58.3 jk
*Cephalosporium acremonium*	EG11M1-1	13.6 k	2.1 lm	84.0 bc
*Exerohilum rostratum*	EG4M1-1	21.0 gh	21.2 d	57.8 jk
	EG4M1-2	23.1 e–g	25.2 b	51.7 l
*Fusarium chlamydosporum*	EG5M1-1	12.6 k	5.0 jk	82.4 c
*Fusarium incarnatum*	EG5M3-1	25.0 de	30.0 a	45.0 m
	EG5M3-2	20.0 hi	10.0 gh	70.0 f
	EG5M3-3	25.0 de	23.3 c	51.7 l
*Fusarium nygami*	EG5M6-1	20.0 hi	22.9 cd	57.1 k
*Fusarium proliferatum*	EG5M4-1	18.5 ij	13.3 f	68.2 fg
	EG5M4-2	26.6 cd	8.3 hi	65.1 gh
	EG5M4-3	33.1 a	6.4 j	60.5 ij
*Fusarium verticillioides*	EG5M5-1	29.8 b	5.0 jk	65.2 gh
	EG5M5-2	25.9 d	17.1 e	57.0 k
	EG5M5-3	25.5 de	3.7 kl	70.8 ef
*Sarocladium zeae*	EG6M2-1	16.2 j	10.2 gh	73.6 de
	EG6M2-2	28.4 bc	11.8 fg	59.8 jk
	EG6M2-3	17.3 j	8.3 hi	74.4 d
	EG6M2-4	23.2 e–g	10.0 gh	66.8 g
Control (without infection)		4.0 m^c^	0.7 m	95.3 a

Values within a column followed by the same letter(s) are not significantly different according to Duncan’s multiple range test (*p* ≤ 0.05).

**Table 2 plants-11-02347-t002:** Pearson moment correlation (r) matrices between five weather variables documented from April to August 2019 in 25 maize-growing governorates of Egypt from which grain samples were gathered (one weather station/governorate).

	Temperature		Relative Humidity		Precipitation	Wind Speed		Solar Radiation
Temperature	1 ^a^							
Relative Humidity	−0.91	***	1					
Precipitation	0.10		−0.22	*	1			
Wind speed	0.15	*	−0.23	*	−0.09	1		
Solar Radiation	0.60	***	−0.70	***	−0.11	0.18	*	1

^a^ Values followed by * and *** are significant at *p* ≤ 0.05 and *p* ≤ 0.001, respectively.

**Table 3 plants-11-02347-t003:** Results of ordination of the canonical correspondence analysis accounted for the first five axes.

Axis	1	2	3	4	5
Eigenvalue	0.120	0.051	0.025	0.012	0.002
Species-environment correlations	0.575	0.496	0.399	0.311	0.155
Cumulative percentage variance of species—weather relation	57.1	81.1	93.0	98.9	100

## Data Availability

Not applicable.

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
