# Peer review of "Distribution and Biodiversity of Seed-Borne Pathogenic and Toxigenic Fungi of Maize in Egypt and Their Correlations with Weather Variables"

_plants, 2022, doi:10.3390/plants11182347_

Round 1

Reviewer 1 Report

Manuscript plants-1904771, " Distribution and biodiversity of seed-borne pathogenic and toxigenic  fungi of maize in Egypt and their correlations with weather variables" is a solid paper with good and useful information. The English used is good overall, except for sparse small errors (including typos), so I have corrected those in the attached version of the paper (please take into account that some errors may have been overlooked or introduced, though). I have also made a few comments in the attached document.

Author Response

Dear Editor-in-Chief 

We would like to send a sincere thanks to our reviewer for thoughtful critiques of our manuscript titled "Distribution and biodiversity of seed-borne pathogenic and toxigenic fungi of maize in Egypt and their correlations with weather variables". Please find the response to your comments and revised version with the tracking changes.

Reviewer 2 Report

Manuscript “plants-1904771” presents a study regarding the biodiversity of pathogenic and toxigenic, seed-borne pathogenic fungi of maize in the main producing regions of Egypt, also analyzing the effect of environmental factors on their occurrence.  Results of the present work provide significant information on the epidemiology of the implicated fungal species, that could be useful in the development of disease management.

The research organization and methodology are well designed and executed, while the results have been properly analyzed, with good quality figures and tables provided. The text is well-written and structured.

However, I have some suggestions that I cite below:

Line 39: There is a need for a reference.

Line 138: “their habit characters” should be rephrased.

Line 160: “was also done” changed to “was also performed”.

Line 161: Please define properly “important fungi”.

Line 173: Authors should provide further information regarding the ML phylogenetic analysis: Phylogeny test, substitution model, rate patterns, interference options, and no. of threads used. In addition, were they any outgroup(s) used?

Paragraph 3.3 (Phylogenetic analysis): (a) Reference species isolates are not included in the phylogenetic tree; (b) the tree seems not to be rooted. Both should be addressed.

Line 512: “Subsequent” should be written in lowercase.

Author Response

(The authors gave the same response as above.)
